# Matrix Wave^TM^ System for Mandibulo-Maxillary Fixation—Just Another Variation on the MMF Theme? Part I: A Review on the Provenance, Evolution and Properties of the System

**DOI:** 10.3390/cmtr18030032

**Published:** 2025-07-12

**Authors:** Carl-Peter Cornelius, Paris Georgios Liokatis, Timothy Doerr, Damir Matic, Stefano Fusetti, Michael Rasse, Nils Claudius Gellrich, Max Heiland, Warren Schubert, Daniel Buchbinder

**Affiliations:** 1Department of Oral and Maxillofacial Surgery, Facial Plastic Surgery, Ludwig-Maximilians University Munich, 80337 Munich, Germany; paris.liokatis@med.uni-muenchen.de; 2Department of Otolaryngology, University of Rochester Medical Center, Rochester, NY 14642, USA; timothy_doerr@urmc.rochester.edu; 3London Plastic Surgery Centre, 123 Dundas St, London, ON N6A 1E8, Canada; dmatic@londonplasticsurgery.ca; 4Department of Maxillofacial Surgery, University of Padova Medical Center, 35122 Padova, Italy; stefano.fusetti@unipd.it; 5Department of Oral and Maxillofacial Surgery, Center of Reconstructive Surgery, University Hospital, Paracelsus Medical University, 5020 Salzburg, Austria; michael.rasse@outlook.at; 6Department of Oral and Maxillofacial Surgery, Hannover Medical School, 30625 Hannover, Germany; 7Charité–Universitätsmedizin Berlin, Corporate Member of Freie Universität Berlin and Humboldt-Universität zu Berlin, 13353 Berlin, Germany; max.heiland@charite.de; 8Division of Plastic Surgery, University of Minnesota, Minneapolis, MN 55455, USA; warrenschubert@aol.com; 9Division of Oral and Maxillofacial Surgery, Department of Otolaryngology, Icahn School of Medicine at Mount Sinai, Mount Sinai Beth Israel, New York, NY 10003, USA; daniel.buchbinder@mountsinai.org

**Keywords:** mandibulo-maxillary fixation (MMF), intermaxillary fixation (IMF), bone anchorage, arch bars, hybrid MMF systems, Matrix Wave^TM^ MMF System

## Abstract

Study design: The advent of the Matrix Wave^TM^ System (Depuy-Synthes)—a bone-anchored Mandibulo-Maxillary Fixation (MMF) System—merits closer consideration because of its peculiarities. Objective: This study alludes to two preliminary stages in the evolution of the Matrix Wave^TM^ MMF System and details its technical and functional features. Results: The Matrix Wave^TM^ System (MWS) is characterized by a smoothed square-shaped Titanium rod profile with a flexible undulating geometry distinct from the flat plate framework in Erich arch bars. Single MWS segments are Omega-shaped and carry a tie-up cleat for interarch linkage to the opposite jaw. The ends at the throughs of each MWS segment are equipped with threaded screw holes to receive locking screws for attachment to underlying mandibular or maxillary bone. An MWS can be partitioned into segments of various length from single Omega-shaped elements over incremental chains of interconnected units up to a horseshoe-shaped bracing of the dental arches. The sinus wave design of each segment allows for stretch, compression and torque movements. So, the entire MWS device can conform to distinctive spatial anatomic relationships. Displaced fragments can be reduced by in-situ-bending of the screw-fixated MWS/Omega segments to obtain accurate realignment of the jaw fragments for the best possible occlusion. Conclusion: The Matrix Wave^TM^ MMF System is an easy-to-apply modular MMF system that can be assembled according to individual demands. Its versatility allows to address most facial fracture scenarios in adults. The option of “omnidirectional” in-situ-bending provides a distinctive feature not found in alternate MMF solutions.

## 1. Introduction

Mandibulo-Maxillary Fixation (MMF)—also referred to as Inter-Maxillary Fixation (IMF)—to immobilize the mandible and maintain occlusal relationships is needed in a number of clinical scenarios, including lost or loose teeth, fractures of the alveolar process, single or multiple fractures of the non-condylar and condylar mandible and/or the maxilla. MMF can be employed for temporary intraoperative use, fragment reduction, repositioning, osteosynthesis, conservative fracture treatment, and prolonged functional treatments. The scope of techniques and devices ranges from simple (Ernst ligatures—1927 [1],1932 [2]; Ivy/Blair eyelets—Ivy 1921 [3], Blair and Ivy 1923 [4]; Risdon cable—Risdon 1929 [5]) through to entangled multiple-loop forms (Stout ligatures—Stout 1943 [6]; and their modifications—Obwegeser 1952 [7]) of direct and indirect interarch dental wiring, prefabricated/commercial tooth borne arch bars, miniscrew hooks (Otten 1981 [8]), conventional or specialized bone screws (Arthur and Berardo 1989 [9]), and hanger plates (Rinehart 1998 [10]). These are all acknowledged MMF appliances and meet many of the required criteria.

Most techniques date back to early concepts (Gilmer 1887 [11], Sauer 1881 [12], 1889 [13]), and long-established variations are still pursued, such as Hauptmeyer- (1917) [14] and Jelenko-type arch bars (Aderer/Jelenko—US Patent No 1,638,006, August 1927 [15]), the Erich arch bar (Erich 1942 [16], Erich and Austin 1944 [17]), or ‘acrylic’ wire splints (Schuchardt 1956 [18], Schuchardt et al., 1961 [19], Stanhope 1969 [20]). Over time, some devices and arch bars have been altered and refined in design, application mode, and material.

Examples include the translation of conventional arch bars into a Titanium alloy with optimized outline for easier application (Iizuka et al., 2006 [21]), a resin bondable version of the Erich arch bar to replace wiring (Baurmash 2006 [22]), an adjustable Nylon “cable tie” device (e.g., Rapid IMF™, Synthes, Paoli, PA, USA), fastened around the neck of a maxillary and mandibular tooth, to provide opposite anchorage points for an interarch elastic power chain (McCaul et al., 2004 [23], Pigadas et al., 2008 [24], Cousin 2009 [25], Johnson 2017 [26]). A later MMF zip-tie concept uses ties put through opposite embrasures for occlusal linkage (i.e., Minne-Ties^®^, Agile MMF, Invisian Medical, Prairie Village, KS, USA) (Jenzer et al., 2022 [27]).

CAD/CAM technologies have been investigated, if 3D printing of patient-specific arch bars is feasible (Druelle et al., 2017 [28], Tache et al., 2021 [29]).

More recent innovations with hybrid or bimodal characteristics include self-made modifications of Erich arch bars for direct bone screw support (de Queiroz 2013 [30], Suresh et al., 2015 [31], Hassan et al., 2018 [32], Rothe et al., 2018 [33]/2019 [34], Pathak et al., 2019 [35], Venugopalan et al., 2020 [36]).

There are commercial products now, including the Stryker SMARTLock^TM^ Hybrid MMF System (Stryker), the OmniMax^TM^ MMF System (Zimmer Biomet), and the Matrix Wave^TM^ MMF System (DePuy-Synthes), which constitute this category of devices. There is another hybrid MMF device patented (US Pat No.10,470,806 B2, 12 November 2019 [37]) and assigned to KLS Martin 2019, which is not available internationally yet.

With reference to two previous evolution stages, this study reviews the technical features of the actual Matrix Wave^TM^ MMF System, showcases the principal application technique and discusses potential unfavorable biomechanical side effects that can occur with indiscriminate vertical plate alignment in relation to the occlusal plane.

## 2. Evolution–Provenance

The commercially available Matrix Wave^TM^ MMF System involved several developmental design and test stages, two of which are explained further.

### 2.1. Locking Adaption Plate and MMF Nuts

The initial idea of a screw-supported arch bar MMF System was based on a line extension of an already-existing locking adaptation (Mini UniLOCK^TM^ 2.0 mm, Synthes, West Chester, PA, USA) plate.

The necessary add-ons would consist of washers serving as spacers to limit the contact and compression of the mucosa on the underside of the plate at the screw sites along with knobs or “nuts” to act as tie-up cleats to accommodate wire ligatures or elastic rubber bands for MMF (Figure 1, Figure 2 and Figure 3).

The washers were cut from an adaptation plate as single ring-shaped pieces.

The use of the locking mechanism between the threaded plate holes and the second large-diameter threads of the specialized screws allowed the plate to standoff from the mucosal soft tissues, preventing their compression.

The solid MMF nut, reminiscent of a mooring bollard, was composed of a threaded bottom to engage in the thread of the plate holes, a conical central part merging into a slight upwards slope and a distinctive rim with a larger diameter at the top. The head of the MMF nuts had a milled star drive recess for self-retainment on the screwdriver and reduction in camming out.

The MMF nut design avoided sharp edges, which could cause irritation and erosion damage of the lip or buccal mucosa

### 2.2. Predecessor—First-Wave Plate Version

Seeking a more practical and simplified solution with fewer components than described for the MMF nuts above and locking adaptation plate as carrier led to the concept of an arch bar in the form of a thin Titanium rod running in a continuous harmonic wave-line of bell-shaped single elements, which provides flexible linkage sites for bone attachments and interarch connections. The turning points (peaks at the crests or troughs, respectively) of the sinusoid wave ‘plate’ prototype were furnished either with hooks or with locking screw-receiving holes along the opposite same phase sides (Figure 4 and Figure 5).

In continuation of the adaptation plate above, the terms wave or snake “Plate” were derived as the first designations for the serpentine hybrid device embodiment. A “Plate” is commonly associated with the conception of a bar-like or baton shape, which is opposed to a wave configuration. So, in a strict sense, “Plate” is somewhat of a misnomer but has become a colloquial and practical term.

Each wave ‘plate’ segment, consisting of a wavelength section and two consecutive plate holes, could be stretched out, squeezed, curved or bent out of plane to place the screw holes between the tooth roots (Figure 4 and Figure 5). This handling will inevitably lead to an irregular three-dimensional formation of the overall plate pathways with dents, bumps and stairsteps.

The alloy and the smoothed rectangular cross-section of the Titanium rod imparted enough malleability, ductility and toughness to allow for extensive manipulation, even for meandering or zig-zag courses. Massive shifts, however, interfere with a long-established premise in MMF usage, since the interfaces with the hooks for interarch occlusal securement require a lineup opposite to each other, at best at the height of the tooth equators (Figure 4 and Figure 5), so mutual interactions during the contouring and bending process need to achieve an appropriate balance of divisions.

This preliminary wave plate version underwent repeated testing on artificial skull models (Figure 5) and in anatomy wet labs on postmortem human subjects to identify its flaws and shortcomings. It was never administered in clinical cases.

### 2.3. Matrix Wave System—Final Design and Technical Description

Since the strictly sinusoidal design and flat rectangular cross-section of the wave plate prototype (Patent No.: US 2011,0152946 A1-23 June 2011 [38]) did not provide enough resilience and residues for easy malleability for the required spatial anatomic adaptation without widespread interactions over several segments during the contouring process, the layout of the wave plate segments was changed into a two-plane (‘split level’) Omega-shaped design (United States Patent, Patent No.: US 9,820,77 B2–23 June 2018 [39]). The plates step off from the screw-receiving locations provided additional clearance for anatomic structures.

The accentuated Omega geometry considerably improved the material reserves for bending, stretching and torque.

Two Matrix Wave embodiments (MWS) make up the framework for MMF (Cornelius and Hardemann 2015 [40]).

A Wave plate—the analogue of an arch bar—consists of a pure Titanium alloy (Ti-6AI-7Nb) rod with a quadrangular (1 × 1 mm) cross-section machined and mechanically manufactured into a continuous row of Omega segments that measures an initial pre-cut length of 11 cm (Figure 6 and Figure 7). The MWP consists of alternating crests and troughs. Their traveling sequence and directions are reversed in consonance with the MWS set-up in one of the jaws matching with vertical mirror images. Along the mandible, the wave line starts with a convexity or peak (cleats open in downward direction) followed by a convexity or valley; in the maxillae, this is the opposite, with the cleats open in the upward direction. The vertices or bottoms of the Matrix Wave plate up- or downsides, respectively, are fitted with a total of nine cleats. The tie-up cleats are geometrically softened and shaped to balance the needs to minimize soft tissue irritation and provide ease of accessibility for cerclages. A total of 10 threaded locking holes are integrated into the MWP, so that a pair of plate holes borders an Omega segment on each end and serves as a basis for a tandem screw fixation of a single segment.

The Matrix Wave plates (MWPs) come in two sizes to conform with the individual anatomy and vertical height of the mandibular/maxillary alveolar rims and dental crowns. The high- and low-profile variants differ in the slope curvatures of the Omega segments (Figure 6 and Figure 7). The smaller MWP profile (“short plate”) results from tighter bends and a stuffed-though more sweeping overall layout. In addition, the plate holes are retained inside the course of the curving line, in contrast to the high profile (“tall plate”), where the plate holes are protruding outside, thus increasing the plate’s vertical height. These varying profiles offer options for patients with smaller anatomy surrounding the dentition.

In transverse profile view, the MWP surface covers two distinct planes (Figure 8). The narrow platform containing the plate hole has a crescent shape and is situated at a lower level (locking hole section) and spaced apart from the raised Omega-loop excursion by a step-off.

The curvatures of the elevated Omega loop offer ample freedom for bending maneuvers, also providing clearance to avoid the dentition and nearby mucosa.

The wing-like short tie-up cleats flanking the vertices of the Omega segments actually take off to a third level. These hooks are 5 mm long, buckling inwards and slightly upwards of the Omega loops. This tie-up cleat styling simplifies wire loops or rubber band placement (Figure 8).

MWPs are fixated with specially designed self-drilling locking screws (MWS/MWP locking screws) made of Titanium alloy (Ti-6Al-7Nb) (Figure 9). These screws have a raised head feature, designed to help minimize soft tissue overgrowth while providing a means to introduce cerclage wire from screw to screw for additional functions, such as reapproximating fragments and bridal wiring.

The screws have a shaft length of 6 mm or 8 mm from the first full winding of the bone threads to the tip. The thread diameter is 1.85 mm, and the core diameter is 1.5 mm. The core diameter tapers towards the tip and decreases stepwise. The cap-shaped screw head, together with the other subdivisions below—the recessed neck, the threaded large-diameter conical locking head for engagement in the plate hole and the neck at its base—have an overall length of 3.8 mm.

In comparison, specialized MMF screws have a center piece underneath their head cap with single or crosswise channels for ligatures with a thread diameter of 2.0 mm and threaded shaft lengths varying between 8 and 16 mm (Cornelius and Ehrenfeld 2010) [41].

A locking screw must be entered centrally into the plate hole to sink the conical locking head at full length. In theory, the MWP locking screw should be adjusted almost perpendicular to the plate. The special locking screws offer variable angle capabilities in any direction. In actual use, the plate/screw locking interface design will allow for an angulation up to 15 degrees.

As the self-drilling screw shaft is turned into the bone and tightened, the wider threads of the conical locking head engage within the threads in the plate hole and keep the plate at a standoff above the mucosal surface.

If the screw is angulated more than 15 degrees, the conical locking head engages prematurely, with some threads remaining seated, more or less above the top surface of the locking hole section of the plate (Figure 10A). In practice, this can result in inadvertently pushing the locking hole section against the mucosa when trying to further tighten the screw. As a safeguard to prevent compression, ischemia and necrosis, an appropriate distance in between the plate and mucosal surface should be maintained by temporarily interposing the blunt tip of a Freer-type elevator as a spacer.

In addition to the plate tie-up cleats, the recessed necks between the screw head and the locking head offer another anchor for the reception of interarch wires or elastic loops (Figure 10A–D).

The bone attachment of MWPs could possibly be accomplished with conventional MMF/IMF screws, if their countersink is conical and larger in diameter then the MWP screw-receiving holes (e.g., 2.0 mm self-drilling, self-tapping IMF screws–Synthes).

The disadvantage of such off-label utilization of incompatible non-locking screws is obvious, however. The MWP is highly likely to be pressed deep into the mucosa so must be firmly rejected.

## 3. Methods

### 3.1. Matrix Wave Plate—Segmentation and Malleability

Prior to assembly, the MWP is provisionally bent, measured and then shortened or split into relevant segments using a specialized plate step cutter or strong cutting pliers. The cuts are made next to a plate hole, leaving its circumference fully intact in order not to weaken the integrity of the ring structure and to protect the locking threads inside. The cut ring exterior is deburred and smoothened. Plate segments of any length must finish in a screw hole on each end to procure secure tandem fixation with two screws. In view of this prerequisite, a full-length MWP (Figure 7 and Figure 8) can be separated up to a maximum of 5 individual Omega-shaped segments (Figure 11).

Wave Plate segments are very flexible and adapt easily to a different intraoral anatomy and their respective requirements.

The ideal plate/screw hole positioning within the interdental tooth roots necessitates bending in an in-plane direction and that means an axial extension or compression of the Omega segments (Figure 12). First adjustments can be made by hand. In accordance with the manufacturer’s specifications, the maximum elongation should not exceed 10 mm from the neutral position between two screw/plate holes. During the plate adaptation using long nose pliers, any deformation of the actual plate holes must be avoided.

Concomitant out-of-plane bending permits additional torsional adjustments, a step-wise or even an overall meandering course of the plate segments. Owing to the Omega pattern, rather extensive alterations and readjustments of the MWP can be effected by spreading, narrowing or unwinding, even after screws have been installed once. It is unnecessary to remove the screws, as they act as anchors in between the pliable Omega segments.

Maximum compression of a one-wave plate element combined with sliding its arms and plate hole in front of each other produces a small-based single hook that can be attached to the bone with a single 8 mm screw passing through the two overlying plate holes (Figure 13 and inset). A set of three or four such hooks in a triangular or quadrangular formation may serve as simple means for a swift MMF (Figure 13), which is functionally similar to the classic self-bent stainless-steel wire hooks fixed with screws to the mandible and maxillae suggested by J.E. Otten for intermaxillary immobilization more than four decades ago (Otten 1981 [8]). This means off-label use.

Even extreme stretching of the plate segments is feasible because of the material and design properties. However, stretching a Wave Plate segment over more than 10 mm is explicitly off-label use (Figure 14). Excessive and back-and-forth bending will impair and fatigue the plate strength.

Despite these material property concerns, two far-stretched Wave Plate elements will provide a rigidly bone anchored arch bar of up to 32 mm (tall-height plate) or 28 mm (small-height plate) length, which can be used to fence in and wire repositioned teeth or the dentition within an alveolar process fracture along the vestibular tooth surfaces (Figure 14).

### 3.2. Matrix Wave^TM^ Plate MMF System—Mode of Application

MWP can offer solutions to plenty of MMF problems. The proposed purpose is for temporary immobilization of mandibular and maxillary fractures intraoperatively during open reduction and internal fixation and the subsequent postoperative period of bone healing (about 6–8 weeks) in patients with permanent dentition.

Inconsistent with this application is the report of MWP use for closed treatment of adult multiple concurrent mandibular fractures (Kiwanuka et al., 2017 [42]).

Currently, the usage of Wave Plates at their full length is promoted by the producers. At the outset, the appropriate plate height is selected, and the full-length plates are contoured to extend completely around the outer surfaces of the mandibular and maxillary arches from molar to molar region, with the open end of the tie-up cleats pointing towards the gums.

Our preference is to use short MWP segments. Whenever it appears appropriate and expedient, just singular MWP segments with a plate-receiving hole on either side of the central Omega arch (Figure 12) for “tandem fixation” are utilized. The pre-/intraoperative transmucosal application of these elements in the premolar/molar transition zone is representative of the procedural technique (Figure 15A–L).

First, the suitability of a sole Omega MWP segment to cross the vertical height along the transition zone of the attached gingiva into the vestibular moveable mucosa is checked. Any interference with the occlusion and jaw movements is avoided by selection of the appropriate plate height and positioning out of the dental articulation pathways. Of course, the vertical height of the plate interacts with the width and will conform to the wingspread of the segments across the tooth roots. When the Wave Plate is initially adapted to the outer contours of the dental curvature or the gingival line, the elements are stretched out or pinched provisionally to span, for instance, over two premolars or a premolar and molar.

The adjusted plate segment is held on the tie-up cleat with a hemostat, and the anterior plate hole is targeted on the mucosa over the bone zone between the tooth roots, which is often perceived by blanching of the juga alveolaria bordering it (Figure 15A). The first screw is turned in through the anterior plate hole no further than to the tapering cone below its locking threads (Figure 15B). Thus, the locking mechanism is not activated yet, which provides further adjustments of the plate and localization of the posterior plate hole. Looking through the free posterior plate hole helps to target the inter-root/interdental insertion point for the posterior screw accurately. This screw is inserted and advanced into the bone (Figure 15C) until shortly before the lowest thread of the conical locking head engages into the plate hole. The plate segment is still loosely fixed and can be lifted up with a thin blunt instrument (e.g., Freer elevator). The screws are then alternately tightened (Figure 15D), so that the plate comes to a firm rest (“standoff”) above the mucosal surface (Figure 15E).

The same steps and protocol are adhered to during the fixation of an Omega segment in the premolar/molar region of the opposite jaw (Figure 15F–J). When all four jaw quadrants are set up with an MWP segment, MMF can be temporarily established, for instance, by a pair of wire ligatures encircling the cleats of the opposing plate units (Figure 15K). This is accomplished prior to or during an open reduction and internal fracture fixation. Postoperatively, the intermaxillary cerclages are released. The plate segments are commonly left in place for an appropriate interval to accept elastics for occlusal guidance or functional treatment, as needed (Figure 15L).

Sections of continuous Omega segments at variable length up to the full extent of the mandibular or maxillary arches may be utilized in a modular fashion. Larger partitions of segments in continuity (≥3 Omegas) can be alternately arranged with shorter sections (=2 Omegas) or singular segments. The array of sections and segments must provide an anatomic interfragmentary adaptation within either jaw if needed and establish and maintain a multi-point stable occlusion between both the anterior and posterior dentition of the jaws (Figure 16A–C). Bridal wires (see below) can also be useful for tight approximation and stabilization of fragments across a fracture line (‘tension banding”).

### 3.3. ‘In-Situ-Bending’ for Fracture Reduction

The extraordinary malleability of the MWP is imparted by its flexible metal rod and the maneuverability provided by its revolving sinus pattern design. These properties make the MWP suitable not only for optimal screw placement but also for fracture reduction. After screw fixation of the MWP plate, conjoined fragments can be efficiently repositioned by molding the Omega segments with a pair of pliers (e.g., the Matrix Wave MMF application instrument), corresponding to an “in-situ-bending” until the interfragmentary gap is readily closed (Figure 17A,B [43,44,45]).

Unlike arch bars, MWPs cannot exercise a tension band function alone. The tension band principle originally means that the tensile forces, in particular in (static) compression plate osteosynthesis along the lower mandibular border, are absorbed to prevent a separation of the fragments at the oral side and contribute to an even load distribution over the fracture site (Spiessl 1989 [46]). If needed, the MWP uses bridal wires as a form of tension banding to reinforce the sagittal and transverse alveolar buttresses in the mandible (Figure 18) and/or the maxillae across the fracture line.

Bridal wire loops are passed around the recessed necks of a pair of MWP MMF locking screws—perpendicular and on each side of the fracture line—to provide stabilization.

Subsequent to “in situ bending” of the MWP for closure of an interfragmentary gap, the bridal loops are tightened to approximate and finally secure the fragments (compare Wang et al., 1998 [47]). Even two soft stainless-steel wires can be twisted together like a Risdon cable wire (Risdon 1929 [5]) and looped around the MWP MMF locking screws located side by side to strengthen the fatigue resistance of tension banding.

## 4. Discussion

The development of this hybrid maxillomandibular system up to the current-state MWP involved several interim stages.

### 4.1. Locking Adaptation Plate and MMF Nuts

The use of a locking adaptation plate in the manner of an arch bar assembled with multiple short MMF “nuts” as cerclage connectors belonged to the hybrid concept from the very start. The plate holes allowed for positioning of bone screws along the plate at spaced intervals, preventing tooth damage by affording “risk-free” holes located over interradicular areas.

The MMF nuts projecting from the outer surface of the plate for interarch connection were placed in the empty plate holes at opposing locations in the upper and lower plate. The spacing between the screws was predefined based on the distance between the holes of the adaptation plate.

Mushroom-shaped buttons soldered to the bucco-labial surface of a conventional arch were suggested long ago (Hasegawa and Leake 1981 [48]). Such a design is more handy than hooks, since it obviates the need to orient arch bars in an upward or downward direction.

Certainly, screwed-in MMF nuts carry the risk of loosening and disengaging from the plate; fortunately, this never occurred in our few clinical cases.

In contrast to conventional arch bars, the locking adaptation plate is more rigid and conforms less exactly to the dentoalveolar relief.

The bone screws locked into the plate build a frame construct as an “external fixator” that rests above the mucosa (Figure 10C) and compensates for minor bending incongruencies.

Placing an arch bar or a plate along or above the gingival margin does carry potential drawbacks. There is evidence that foreign bodies like wire ligatures, obstruction of the gingival sulcus and compromised oral hygiene induce an accumulation of bacteria, plaque formation and inflammatory infiltration of the periodontal soft tissues (Kornmann et al., 1981 [49], Tatakis and Kumar 2005 [50]).

Historically, this resulted in the nominal requirement to adapt and secure the arch bars at the height of the tooth crown equators. To hold that position and make it retentive using preformed commercial arch bars was rarely durable in practice, despite the use of different cross-sections or diameters and attempts to cover the ligature wires with cold cure auto-polymerizing acrylic resin as a safeguard against slippage (Schuchardt 1956 [18]).

Moreover, the vertical level where the fasteners for the intermaxillary connection (with different naming, such as buttons, knobs, nuts, hooks, winglets, cleats, tangs, prongs, etc.) are situated has fundamental biomechanical implications.

In short, from a clinical standpoint, the alignment of the intermaxillary connectors along the tooth equators offers considerable benefit in the majority of facial fractures, because it helps to maintain the re-established occlusal contacts and prevents the risk of splaying going unnoticed on the lingual side (compare Pedemonte et al., 2019 [51] Appendix A eContent, for a detailed explanation, see Appendix A, including Appendix A).

The hybrid MMF prototype solution using a locking adaptation plate, locking screws underlaid with washers (all from the shelves) and MMF nuts imparted valuable insights and highlighted the technical requirements needed to meet the project demands:Adaptability of screw attachment location to prevent tooth root damage by screw;Insertion;Protection against compression of the buccal gingiva/attached mucosa;Vertical apposition of the plate/arch bar at a level non-irritant and atraumatic to periodontal tissues conjointly with the prevention of undesirable biomechanical effects;Flexible selection of interarch connection points (i.e., ‘wire hooks’);Reduction/adjustment of fragments;Tension band function in osteosynthesis of mandibular fractures;Potential advantages of a modular system.

### 4.2. Predecessor—First-Wave Plate Version

The next design idea was a somewhat unorthodox prototype (Figure 4 and Figure 5) as it deviated from the usual longitudinal beam or band-like core element of Erich or Schuchardt arch bars. The wavy design of the Titanium rod bent in successive regular up and downward curves prompted the internal designation of ‘Snake plate’.

This nickname referred not only to the wave-line shape of the device but to its bending properties, resembling two modes of the limbless forward locomotion of snakes—lateral undulation and concertina movement (Gray 1946 [52], Jayne 2020 [53]). In these patterns, movement is accomplished by alternate lateral flexures to the left and right along the length of the snake body, resulting in an undulary propulsion or a series of moving waves; concertina movement is performed via alternate phases of stasis and movement synchronized over several structural body segments. Motion occurs either by pushing (straightening) or pulling an elastic section into an anterior direction, while the successive or the anteceding section is immobilized.

The screw-receiving plate holes at the turning points of the rod could be utilized as a sighting telescope to target low-risk points for bone screw insertion in interradicular and subapical or supra-apical zones, respectively.

However, manipulation to adjust a Wave Plate segment in plane or even more out of plane to bring a screw-receiving hole into its target propagated excursions and deflections into the adjacent segments leads to more or less pronounced changes in their shapes, vertical height and resting positions relative to each other (Figure 5). The same was true for the alignment of the wire hooks (‘tie-up cleats’). To control and compensate for this complex reciprocal and interacting behavior by appropriate counter-bending reactions are somewhat intuitive but will finally end up with a coherent and precisely fitting 3D geometry of the whole Wave Plate. The deep and distant semicircular arches of the first Wave Plate version were recognized to be causative for the exceedingly marked “chain reactions” over its spatial extent.

### 4.3. Second Wave Plate Version: Matrix Wave^TM^ MMF System

The major shortcoming of the first Wave Plate version—too far down- or up-reaching bell-shaped curves—was answered with substantial reengineering. The unfavorable bell-shaped curves were converted into the configuration of a circle broken up at the bottom side, just like the capital letter Omega.

The Omega shape yielded a larger utilizable circumference of the curvatures and increased their malleability without producing adverse side effects on the neighboring plate segments to the magnitude described before.

This “matured” design feature simplified the MWP application to individual patient anatomy—the vertical level of the MWP with its wire hooks and screw-receiving holes became accurately adjustable over the entire range of the maxillary and mandibular dento-alveolar arches.

The Omega design also complemented and refined the “in situ bending” capabilities of the MWP during fracture reduction.

The serpentine-inspired MWP design and its peculiarities give good reasons to survey some aspects relevant to screw-based MMF applications and conventional Erich arch bars.

In discussing “hybrid mandibulo-maxillary fixation”, it is necessary to define this as an MMF device, such as an arch bar or a suitable analogue coming in pairs, each of which is fixed with bone screws instead of applying the term to a mixture of conventional arch bars in the maxillae together with skeletal anchoring screws in the mandible (Park et al., 2013 [54]), though this is not erroneous in the strict sense of the word “hybrid”.

De facto new ideas and concepts to improve the versatility, the efficacy as well as the patient’s and surgeon’s safety of MMF have been explored ever since.

The categorial predecessors of the first hybrid MMF devices (de Queiroz 2013 [30]) were the classic tooth-ligated Erich arch bars (EABs) (Erich 1942 [16], Erich and Austin 1944 [17]) and bone screws in different varieties from conventional cortical bone screws at the beginning (Arthur and Berardo 1989 [9]) to specialized self-drilling and/or self-tapping bone screws with various modifications (e.g., channels) of their screw heads to ease the wrapping or passing of intermaxillary cerclage wires, alleviating compression of the mucosa surrounding their osseous entrance spots and to pad the overlying buccal soft tissues against painful erosions and pressure ulcerations.

Longitudinal beam or band-like arch bars provided connection points for cerclage wiring or elastic loops by attached wire hooks, cleats, tangs or similar.

Often reported disadvantages of the classic EABs are their tedious handling and lengthy application, risk of wire stick injuries and transmission of viral pathogens during the application procedures, compromised oral hygiene, periodontal concerns in long-term use, instability of fixation, exertion of orthodontic forces (lateral and vertical extrusion), pain and discomfort perception of the patient, annoying removal and their uselessness in trauma patients with dentofacial deformities such as severe forms of mandibular retrusion (Angle Class III), anterior open or deep frontal overbite (Falci et al., 2015 [55]). Difficulties were also preprogrammed in advanced periodontal disease, with a loose residual dentition that cannot carry an arch bar.

Conversely, bone screw anchorage increased the MMF application in speed and efficacy, decreased the frequency of puncture injuries and facilitated oral hygiene; however, this presented a novel, previously unknown risk in MMF, namely tooth root injuries, since the screws were inserted into the mandibular and maxillary alveolar processes in direct proximity to the dental root tips (i.e., sub- or supra-apical) at first (Schneider et al., 2000 [56], Ho et al., 2000 [57], Maurer et al., 2002 [58], Hoffmann et al., 2003 [59], Fabbroni et al., 2004 [60], Roccia et al., 2005 [61], Imazawa et al., 2006 [62], Coletti et al., 2007 [63]).

Further criticism referred to the proliferation of granulation tissue with overgrowth of the screw heads in the case of placement into the mobile vestibular mucosa and the interference of intermaxillary wire loops with the edges of upper incisors or the canine facettes, lesions of inferior alveolar nerves, maxillary antrum penetration potentially inducing sinusitis, hardware failures, deficient long-term stability (Kauke et al., 2018 [64]) and even fracture propagation through the wedge effect of self-drilling screws (Mostoufi et al., 2020 [65]).

Countless publications and comparative studies addressed many of the aforementioned pros and cons of the tooth-borne and/or bone-borne MMF modalities. These have included prospective (Fabbroni et al., 2004 [60], Nandini et al., 2011 [66], Satish et al., 2014 [67], Karthick et al., 2017 [68], Kumar et al., 2018 [69]) and randomized controlled trials (Rai et al., 2011 [70], van den Bergh et al., 2015 [71], Sandhu et al., 2018 [72], Fernandes et al., 2023 [73]) or appeared pooled in a number of reviews (Alves et al., 2012 [74], Delber-Dupas et al., 2013 [75], Falci et al., 2015 [54], Qureshi et al., 2016 [76], Kopp et al., 2016 [77]) and meta-analyses (Fernandes et al., 2021 [78]).

Due to suspected bias and arguable quality (evidence-level) of the analyzed studies, comparisons between Erich arch bars (EABs) and MMF screws were summarized by Fernandes et al., 2021 [78] as follows:No scientific evidence for differences in stability of the anchorage, restitution ofpreinjury occlusal relationships and contacts as well as patient’s quality of life.MMF screws—shorter operating times at application and removal.EABs—lower risk for iatrogenic tooth injuries, greater risk for skin punctures/wirestick injuries.

In particular, with a view to trauma in the mandible, Park et al., (2023) [79] considered simple fractures with full dentition as ideal indications for MMF screw use in distinction to complex fracture patterns, where EABs would be a better treatment option.

The proposition to prevent wire stick injuries during MMF procedures with reusable custom-made thermoplastic guards for each finger may be ranked as an oddity (Kumaresan et al., 2014 [80]). The application of pre-cut wires via laser welding with rounded blunt ends represents a more readily obtainable mode of protection in MMF wiring techniques, either during the installation of intermaxillary wire cerclages or bridal wires (Brandtner 2015 [81]).

As pointed out initially, hybrid arch bars are an amalgamation between the best features of conventional Erich arch bars and bone fixation screws.

Both modalities have well-known advantages and drawbacks—rigidly embracing the dental arch (inappropriately labeled as—“tension banding”) and slow application versus risk of irreversible tooth root injuries.

Hybrid arch bars have been optimistically touted to incorporate the advantages and to overcome the drawbacks of conventional arch bars and MMF screws (Kendrick 2016 [82], Kiwanuka et al., 2017 [42], Roeder et al., 2018 [83], Ali and Graham 2020 [84], Venugopalan et al., 2020 [36], Sankar et al., 2023 [85]).

Currently, two lines of development have been emerging: self-made or chairside produced modified hybrid arch bars, originating from the modification of an Erich bar (de Queiroz 2013 [30]) and a league of commercially available readymade hybrid MMF systems.

Independent of their manufacturing styles and workmanship, the hybrid appliances still face a tradeoff between speed or operative time savings and the potential risk of tooth root injuries.

The targeting function of the screw-receiving (bone anchor) holes in the bone-bearing structures of the hybrid MMF devices represents a design feature that may contribute to reducing the incidence of tooth injuries due to interradicular screw placement.

With respect to the targeting functionality, the slender framework and serpentine embodiment of the MWP assume a special role, setting it apart from the design of the other commercially available hybrid systems.

This topic and the current state of knowledge on application procedures and clinical outcomes are the subject of a subsequent narrative review and analysis (Cornelius et al., 2025, Part II [86]).

## Figures and Tables

**Figure 1 cmtr-18-00032-f001:**
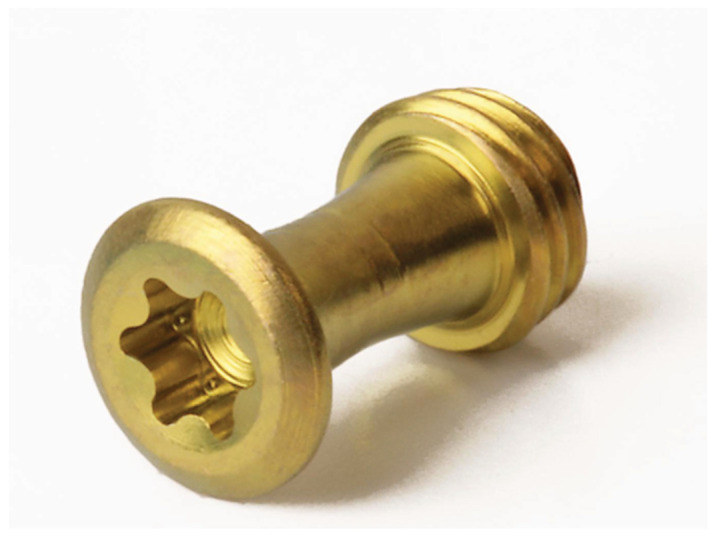
MMF nut–CAM–Titanium workpiece with 6-point star drive (=hexalobular internal) recess head profile. Source/origin: Photograph collection—C.P. Cornelius. Note: MMF Nuts have been custom made (Stratec Medial^®^ GmbH Oberdorf, Switzerland, 2006, REF SM 205300) according to the idea and sketches of the first author (they have not been submitted for patent application).

**Figure 2 cmtr-18-00032-f002:**
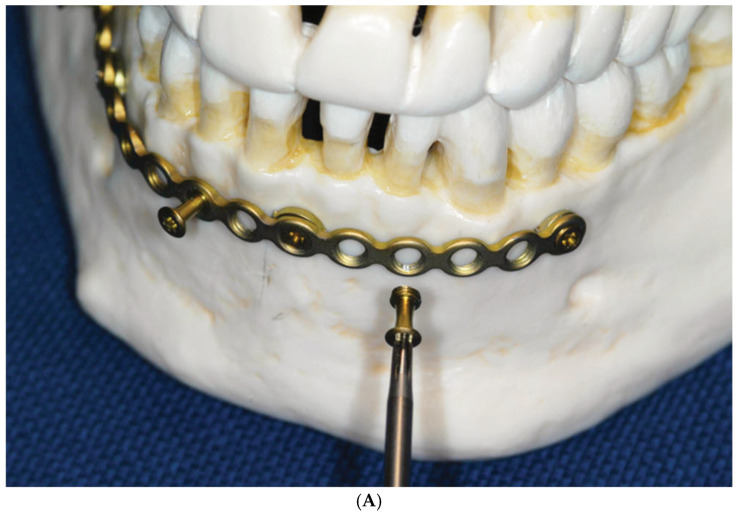
(**A**)**:** Insertion of an MMF nut into Locking adaptation plate. (**B**): Opposing locking adaptation plates in the mandible and maxillae underlaid with single hole (‘ring’) washers beneath the screw sites to keep the plate at a distance (“stand off”) to the mucosa/bone surface and affixed MMF nuts. Source/origin: Plastic skull model, Photograph collection—C.P. Cornelius.

**Figure 3 cmtr-18-00032-f003:**
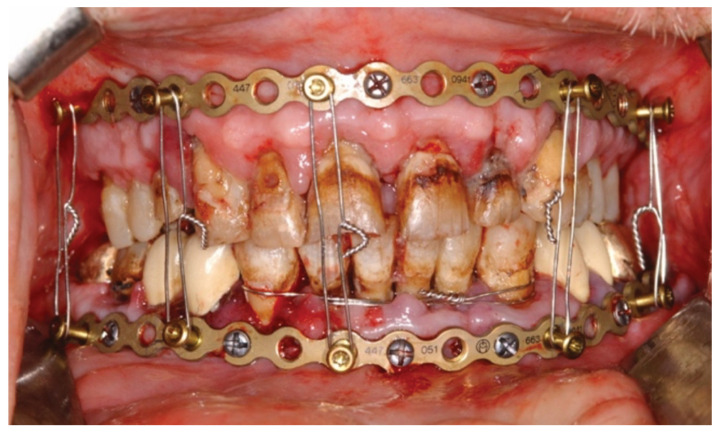
Case Example—Neglected dentition unsuitable for circumdental wire fixation of arch bars in a self-inflicted gunshot injury. Use of an alternative to conventional arch bars: locking adaptation plates secured with self-tapping locking bone screws into the alveolar rims, mounted with MMF nuts and interconnecting wire ligatures hooked over the protruding MMF nuts. Notice: bone fixation with screws inserted along mucogingival junction zones. Source/origin: Photograph collection—C.P. Cornelius.

**Figure 4 cmtr-18-00032-f004:**
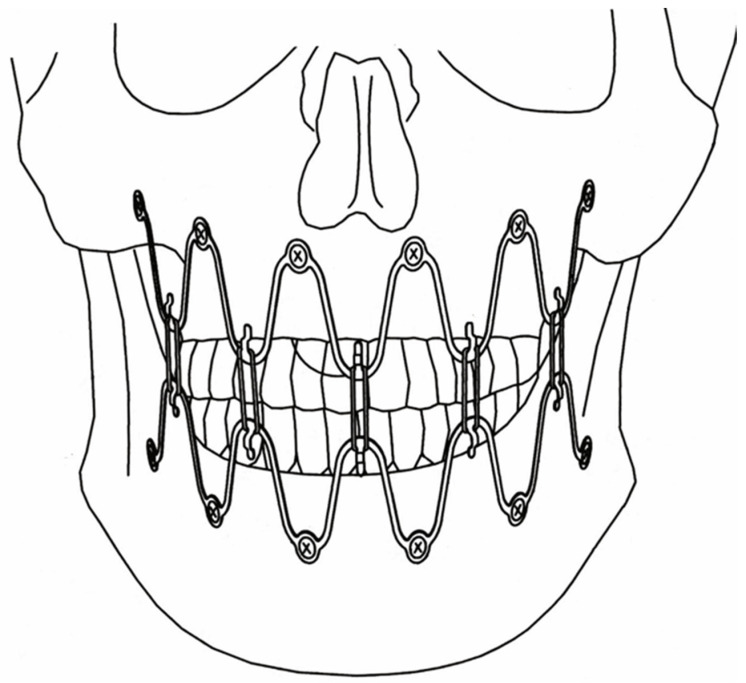
Initial schematic draft of basic concept of a wave-line MMF device—A pair of full arch 5-segment wave “plates” installed in the mandible and maxillae oriented diametrically opposed to each other in a harmonic upwardly and downwardly alternating oscillating pattern of crests and troughs. The interfaces and hooks for interarch/intermaxillary connection relate vis à vis at the level of the tooth equators. The plate holes for bone fixation are intended to lie outside the dentition, below or above, the alveolar processes with their underlying tooth roots (modified Figure 1 from United States Patent, Patent No.: US 2011,0152946 A1–23 June 2011 and Figure 1 from United States Patent, Patent No.: US 10,130,404 B2 2018–20 November 2018). Source/origin: US Patent, Patent No.: US 2011,0152946 A1–23 June 2011 [37].

**Figure 5 cmtr-18-00032-f005:**
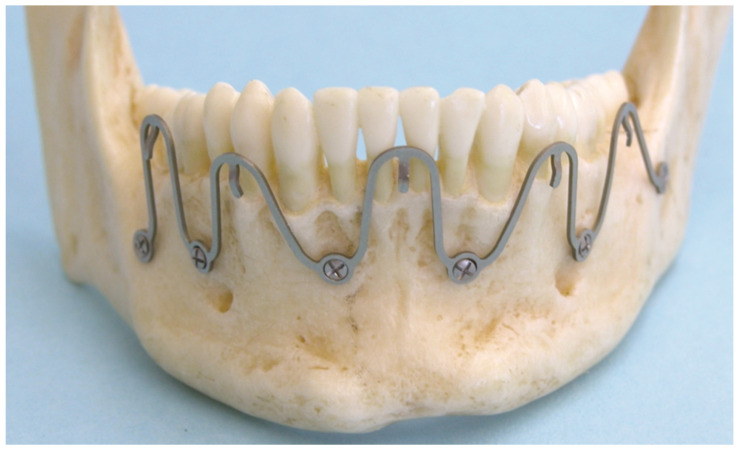
Five-segment wave “plate” prototype mounted on a plastic mandible model. Note: bell curve shape of single wave segments. Median wave segment bell shape unaltered with screw fixation sites between interradicular spaces of lateral incisors and canines. Paramedian and posterior left body wave segments stretched out to reach into subapical zone for screw placement. Posterior right body segment compressed to omit tooth root contact. Wave “plate” crests with wire hooks at slightly varying height adjacent to tooth equator levels. Source/origin: Photograph collection—C.P. Cornelius.

**Figure 6 cmtr-18-00032-f006:**
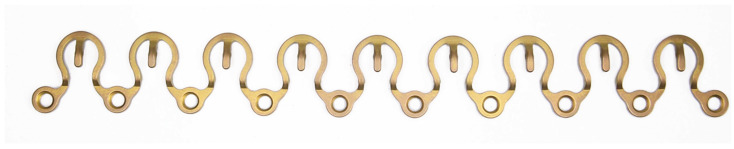
Matrix Wave ™ Plate—high-profile/tall-height (1.3 cm) variant. Entire pre-cut length consists of 9 Omega-shaped segments with 10 plate holes. Alignment of plate holes outside the plate’s curvature line (see text for more details). Source/origin: Photograph collection—C.P. Cornelius.

**Figure 7 cmtr-18-00032-f007:**
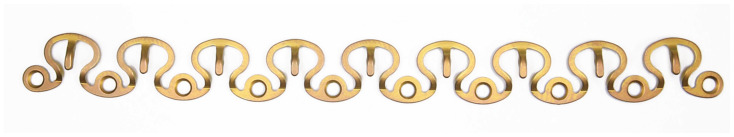
Matrix Wave ™ Plate—low-profile/short-height (1.0 cm) variant, entire pre-cut length. The Omega shape of the elements is more pronounced. Plate holes are bypassing along the inside of the plate’s curvature line (see text for more details). Source/origin: Photograph collection—C.P. Cornelius.

**Figure 8 cmtr-18-00032-f008:**
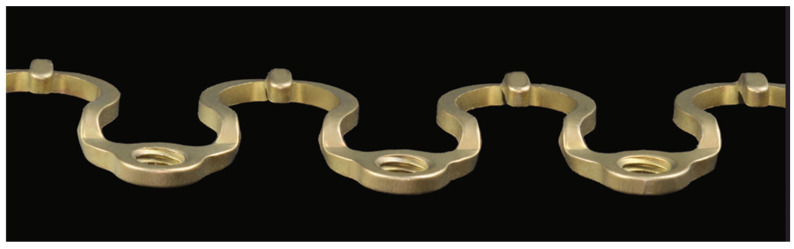
Transverse MWP profile–Horizontal view, revealing the tiered-level arrangement of the plate. Source/origin: Photograph collection—C.P. Cornelius.

**Figure 9 cmtr-18-00032-f009:**
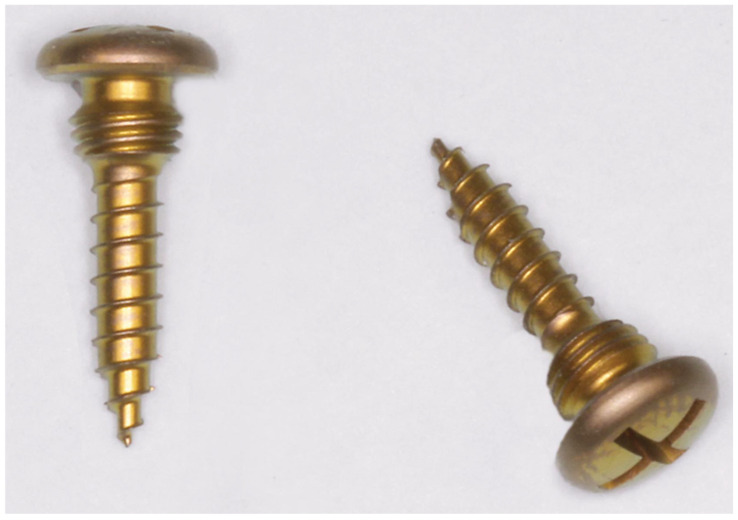
MWP locking screws with threads for plate and threads for bone fixation—short version with 6 mm shaft length from different angles. Note: self-retaining cruciform screwdriver recess (=CMF Matrix Drive-Depuy Synthes). The subdivisions at the top end sum up to 3.8 mm in length. The four subdivisions allot for as follows: cap-shaped screw head, 1.3 mm; recessed neck, 1 mm; threaded large-diameter conical locking head and its neck, 1.5 mm. Source/origin: Photograph collection—C.P. Cornelius.

**Figure 10 cmtr-18-00032-f010:**
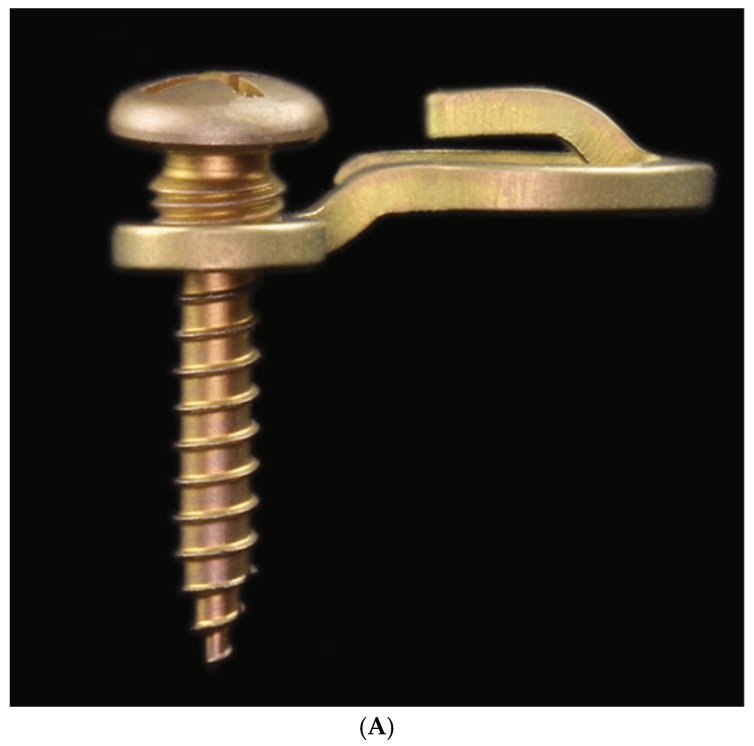
(**A**) Lateral view of a high-profile MWP–MMF locking screw/8 mm shaft length placed axially into center of the plate hole at the locking hole section of the plate. The conical locking head turned into the plate hole showing locking threads above the top surface of the plate. Source/origin: Photograph collection—C.P. Cornelius. (**B**) High-profile MWP—side view showing several plate levels in series. MWP MMF locking screw on axis inside a plate hole with locking threads only partially driven into the plate. Plate tie-up cleats and screw head/neck assume about the same level [39]. Source/origin: Photograph collection—C.P. Cornelius. (**C**) Technical drawing—Isolated MWP locking hole section in cutaway view (Indian yellow) with adjoining plate arms (light yellow). Locking threads of MWP locking screw (shaft length: 6 mm) sunken in all the way down the plate hole and set flush with the surface. In this manner, the potential compression zone between the plate underside and the mucosa tissue layer (red-pink) along the transgingival part of the screw is reliably kept open (red/white double arrows). Length of conical locking head = 1.0 mm; basal neck of locking head = 0.5 mm; not fully threaded start of endosseous screw part = 0.5 mm; length of the true endosseous screw part = 6 mm. In sum, a total length of 7 mm is sticking out at the bottom of the plate. Recessed neck of the screw upper part (height 1 mm—blue arrows) is conveniently accessible for wire or rubber loops. Overall length of a “6 mm” = 10.3 mm and of a “9 mm” = 13.3 mm Source/origin: modified from United States Patent, Patent No.: US 9,820,77 B2–23 June 2018 [39]. (**D**) High-profile MWP, inclined top-down view to visualize the boundaries, crescent shape and spatial extent of the locking hole section from above. Source/origin: Photograph collection—C.P. Cornelius.

**Figure 11 cmtr-18-00032-f011:**
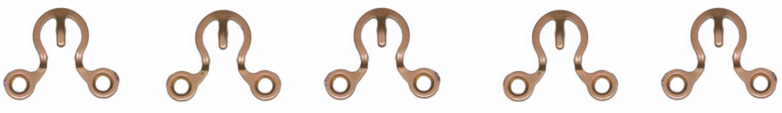
Full-length high-profile MWP cut into 5 individual Omega segments. Each segment with plate holes as terminals at each end. The 4 interpolating mere Omega-shaped excursions dismantled of plate holes for bone attachment have been removed. Source/origin: Photograph collection—C.P. Cornelius.

**Figure 12 cmtr-18-00032-f012:**
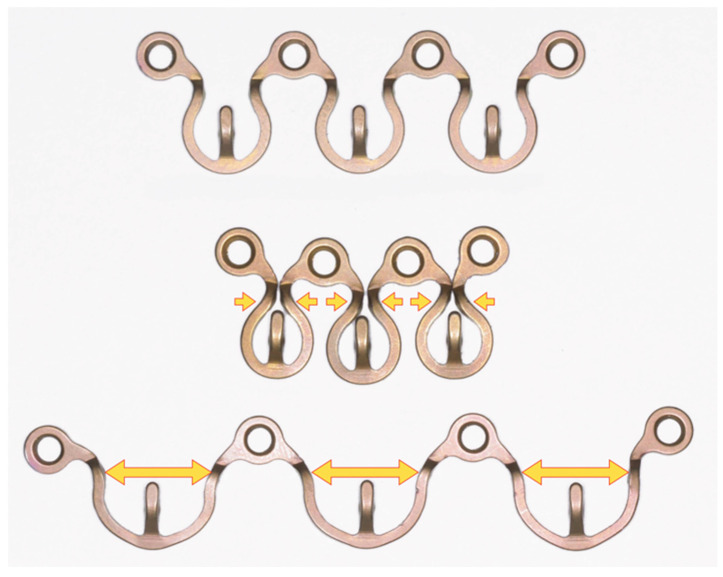
Plate and screw hole positions need to be variable—Omega-shaped MWP segments in neutral length (**above**) are compressible (**middle**) and stretchable (**below**). Source/origin: Photograph collection—C.P. Cornelius.

**Figure 13 cmtr-18-00032-f013:**
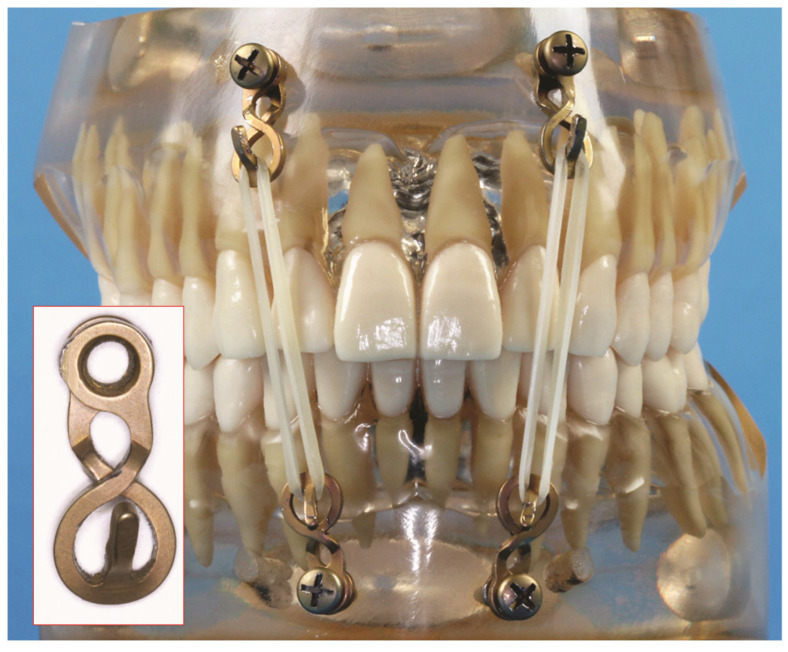
A one-wave plate Omega segment can be easily squeezed into a small-based single-screw-anchored device analogous to an Otten hook (see inset). Dental model with rectangular array of four such singular-screw-fixated Omega segment hooks—MMF by means of elastic loops. Source/origin: Photograph collection—C.P. Cornelius.

**Figure 14 cmtr-18-00032-f014:**
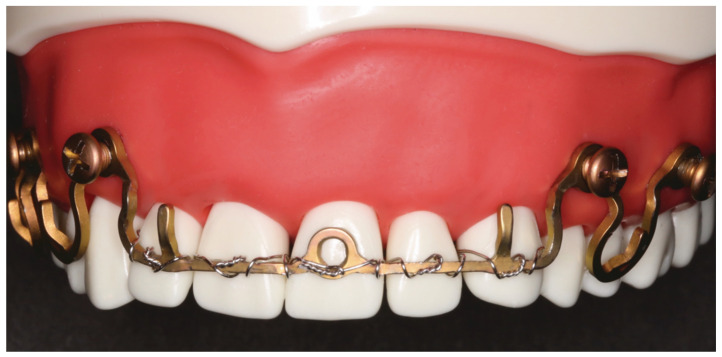
Upper jaw model—Application of two adjacent MWP segments (high profile/tall height) bent open and flattened out as bone anchored arch bar such as in a transverse alveolar process fracture or tooth avulsions—teeth 12, 11, 21, 22 and 23 immobilized by separate wire loops around the MWP bar. Note: This is off-label use! Source/origin: Photograph collection—C.P. Cornelius.

**Figure 15 cmtr-18-00032-f015:**
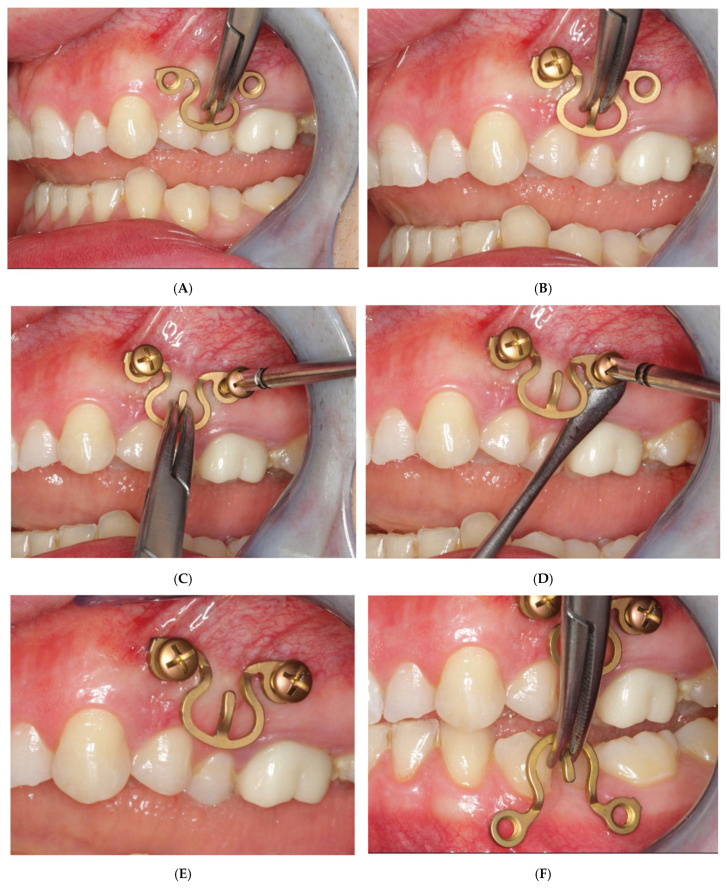
(**A**) Case with a double mandibular fracture—right mandibular angle; left condylar base (see panoramic X-ray in Figure 15L). Deep bite not suitable for conventional tooth borne arch bars. A low-profile MWP Omega segment is tried on along the mucogingival transition zone for interradicular screw fixation in the premolar region of the left maxilla. Interferences with the habitual occlusal position and/or articulatory movements must be ruled out. (**B**) Owing to the more comfortable access, the anterior screw is inserted first. Angulation of the screw should be avoided. The conical locking head of the screw should not get engaged into the plate hole prematurely. (**C**) The posterior screw is turned likewise halfway into the bone for a loose prefixation of the MWP segment. (**D**) The MWP segment is supported with a Freer elevator as a spacer below the flatbed portion, and the screws are tightened alternately until the plate segment is firmly gripped by the locking threads. (**E**) Screw fixation of MWP segment completed. Most of the time, the conical locking head cannot be fully countersunk in the plate hole. It is essential, however, that the threads of the plate hole and the locking threads of the screw effectively purchase. (**F**) High-profile MWP Omega segment adapted to the interradicular spaces in the premolar region of the opposite jaw, in this case, the left mandible. Interferences with the habitual occlusal position and/or articulatory movements must be ruled out. (**G**) Prefixation—plate segment still moveable. (**H**) Posterior screw turned into the plate hole, supporting the plate with a Freer elevator from underneath. (**I**) Anterior screw turned in, while plate segment is supported with Freer elevator—alternate tightening of the screws. (**J**) MWP segment finally fixed in juxtaposition to the vestibular tooth crowns just below occlusal plane. A steep canine guidance, however, averts disruptive contacts between the second upper premolar and the top rail of the plate. (**K**) MWP Omega segments mounted in all jaw quadrants for temporary intraoperative MMF with wire ligatures to immobilize the mandible. Note: Conical locking heads are partially countersunk, only. (**L**) Postoperative Panoramic X-ray after transoral ORIF (miniplate osteosynthesis). Four MWP segments left in place for optional functional treatment during follow-up. All screws for MWP attachment located in interradicular alveolar bone. Inset: four MWP Omega segments as oriented and used in this illustrative case. Source/origin: Photograph collection—C.P. Cornelius.

**Figure 16 cmtr-18-00032-f016:**
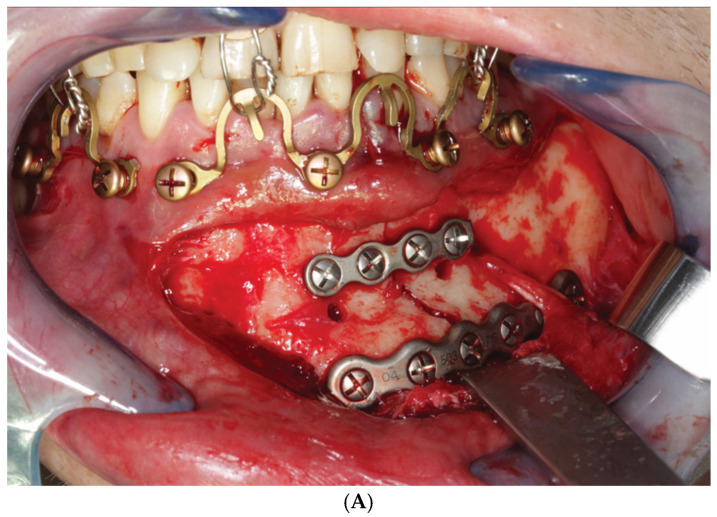
(**A**) Case example—Mandibular double fracture: right ramus and left parasymphysis. Intraoperative view—MWP assembly for immobilization of the lower jaw composed of 3 separate singular MWP Omega segments in the upper jaw opposite to triple Omega segments in a continuous row spanning the parasymphyseal mandibular fracture line and a single Omega segment in the right canine premolar region of the mandible; transoral anterolateral approach for parasymphyseal two-miniplate osteosynthesis. Note: superior 4-hole miniplate acts as a tension band, so that the lower MWP segments must not perform this function. (**B**) (Case example—Mandibular double fracture, cont’d) Transoral miniplate fixation of oblique fracture of right mandibular ramus. (**C**) (Case example—Mandibular double fracture, cont’d) Postoperative 3D CT scan—profile of right facial skeleton elucidating the oblique fracture line course fracture of right ramus and the position of the transoral miniplate fixation. Source/origin: Photograph collection—C.P. Cornelius.

**Figure 17 cmtr-18-00032-f017:**
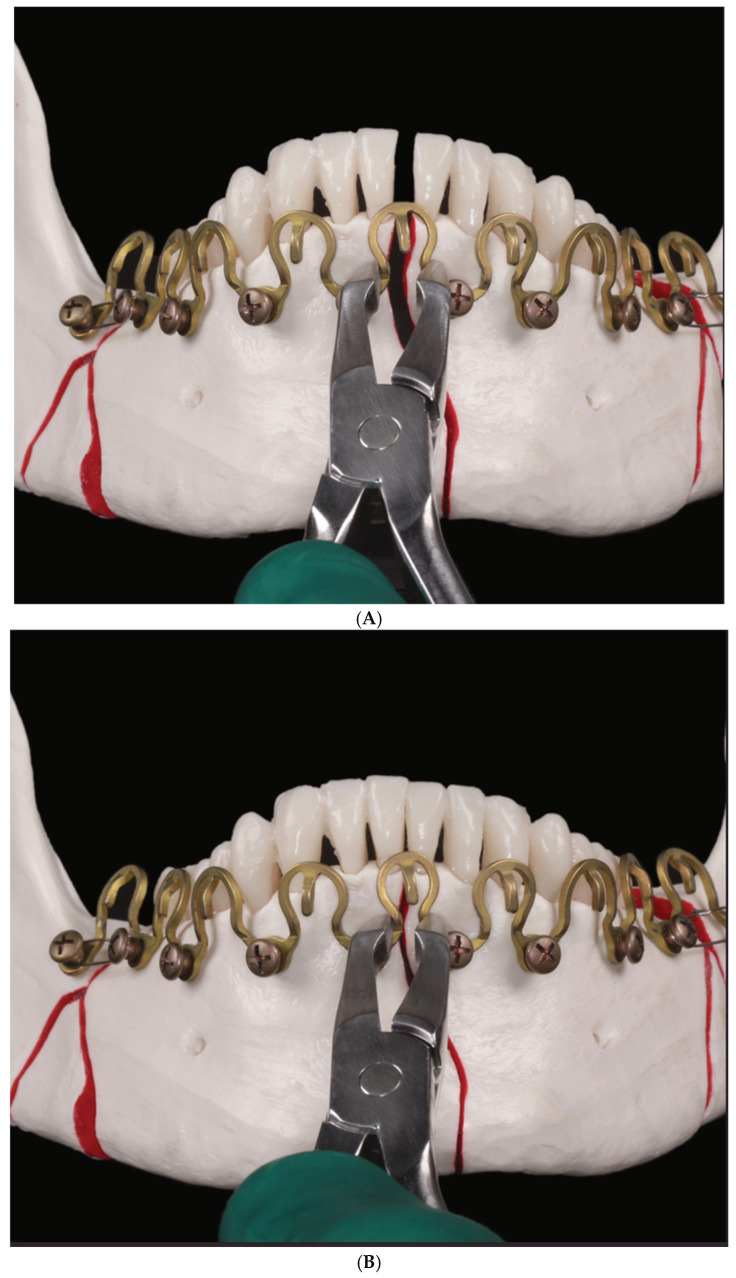
(**A**) ‘In-situ-Bending’—Median (symphyseal) mandibular fracture. Fragments coupled via several segments of a full arch high-profile MWP. Fracture gap still wide open. Pair of pliers about to squeeze the central Omega segment for interfragmentary closure. (**B**) “In-situ-Bending” —Completed. Accurate realignment midway between the two adjacent symphyseal fragments. Where appropriate, further Omega segments nearby can be reshaped to obtain the optimal fracture reduction. Source/Origin: Photograph collection—C.P. Cornelius. (For a cautionary note to Figure 17, see Appendix A).

**Figure 18 cmtr-18-00032-f018:**
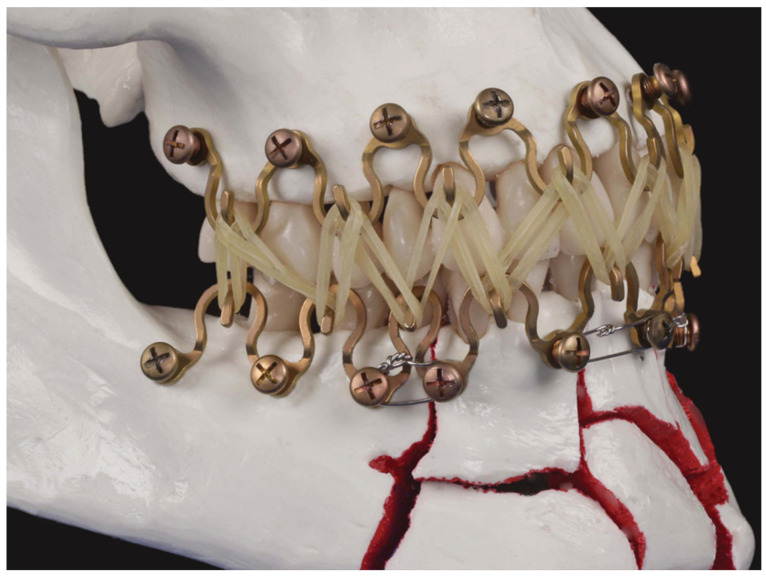
Bridal wires across a fracture line after in situ bending in the right premolar region and for anterior interfragmentary connection. Note: tension banding is necessary for interfragmentary linkage only and not within a fragment. Source/Origin: Photograph collection—C.P. Cornelius.

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
