# Peer review of "Matrix WaveTM System for Mandibulo-Maxillary Fixation—Just Another Variation on the MMF Theme? Part I: A Review on the Provenance, Evolution and Properties of the System"

_1943-3883, 2025, doi:10.3390/cmtr18030032_

Round 1
Reviewer 1 Report
Comments and Suggestions for Authors
Reviews for:
Matrix wave system for MMF – just another variation on the MMF theme?
Part I: A review on the provenance, evolution and properties of the system, and
Part II: In context to self-made hybrid Erich arch bars and commercial hybrid MMF systems – Literature review and analysis of design features
I have combined my reviews for both manuscripts as they are intimately related and many comments relate to both, plus they both have the same authors
Initially and then on review, I was somewhat overwhelmed by the size of each article, at 42 and 67 pages respectively, with a further 16 pages of supplement, amounting to 125 pages in total, this seemed to be more a chapter of or even a textbook in its own right
The overall concept and design of both papers is interesting and informative, although they do not necessarily present any new information, there is deep analysis of the design of several systems (esp. the Matrix Wave Plate – MWP - in Part I) and more than enough information on which the reader can make an informed choice on which system to choose, when and for what
Further, there is enough information, and some, to analyse the performance, the pros and cons of each system with reference to the available literature. The limitations, variations, inconsistencies and omissions of many relevant studies are compared and contrasted/analysed in detail
In particular, the section on dividing and sectioning pieces or sections of the MWP to use in different parts of each arch was extremely useful and instructional. Further, the introduction, discussion and conclusion sections of each part are also well written and contain the essential messages for readers to take away
So, in summary, these are useful and helpful in the context of the available literature, however, crystal clear messages are difficult to obtain from the main body of each manuscript and there are many points that must be considered in terms of critique:
- Parts I and II could be significantly condensed into 1 whole article entitled: A comparative literature review on variations of MMF systems, with specific focus on the Matrix Wave System – this will improve accessibility for readers and not discourage them with 2 large parts, a supplement and 2 long titles
- I suggest that parts I and II are reversed so that the comparison of the available systems is presented 1st and then the 2nd part concentrates on the when, how and why of the MWP system and where it may be superior to its MMF counterparts
- A thorough and forensic English language, grammar and tense review/proof read is required to make these manuscripts more concise and remove Americanised English plus improve sentence structure; many sections in both papers are too long, overly detailed, repetitive and verbose. In their current form, I have no doubt that readers will be disincentivised and demotivated by the amount of detailed technical language in each paper. The authors even allude to this point in their own conclusion at the end of the current part II (lines 1937-40). It is asking an enormous amount of readers to expect them to concentrate on these articles alone for over 100 pages
- Simple diagrams of each system are all that are required, with small technical descriptions. If the authors wish to include system minutiae/technical differences/specifications and theoretical geometric representations, then they can, but they need to be in electronic supplement/appendix form, which readers can reference/read if they wish – I suggest this is done via a QR link to the CMTR website to avoid a large paper volume
- There is no detailed mention of the use of custom made arch bars from internal laboratories and yet these are still in use. These can have longer, slender cleats that are more user friendly than the small cleats on EABs or hybrid versions thereof and may result in a different set of comparative data. The MWP cleat design is very similar to the custom made arch bar type of cleat, and there is no doubt in practice that it is easier to place wire and elastics around MWP cleats than the cleats of the other commercially available EAB/Hybrid types, further, the smooth screw head caps can also be used as an extra bonus
- There is no mention of the risk of and complication of MMF screw (alone or in combination with hybrid arch bars) head fractures, which are known/experienced and can result in retained screw threads
- There is no mention of the type of screwdriver used for MMF devices – be that a press/friction fit or a sleeve type screwdriver – this is relevant because in clinical practice, although slightly bulky the sleeve type results in less disconnection of the screw especially in difficult to access areas in the oral cavity
- Although tooth/teeth injury as a consequence of/risk of screw placement is mentioned repeatedly, there is no evidence presented on the need for, or incidence of, root canal treatment/extractions as a consequence of these iatrogenic injuries, plus they indicate that they will often go on to heal. Neither is it indicated if the teeth injuries were different between different experience levels of operator
- If there is a concern about teeth injuries, then why haven’t custom made hybrid arch bars become part of routine maxillofacial trauma practice? So that screw positions can be guided from the outset to avoid roots? Especially, as were now in the age of 3D/computer planning, with widespread availability and access to CT scans and planning software
- Even though wire stick injury is a risk and is obviously important, there is no evidence presented in terms of the risk/incidence of Blood Born Virus infection (BBV) as a direct consequence of wire stick injury. Indeed, it could be theorised that as wires are solid (not hollow like needles), that the risk of transmission of BBV and tissue should be extremely low. This point refers to part II line 1363
- In terms of very specific corrections that are required:
- Through part I and II the following words would benefit from change, to increase access for a broad readership:
anatomical (not anatomic), anaesthesia (not anaesthesia), trauma (not traumatization), emphasised (not emphasized), mouldable (not moldable), bridle (not bridal), bevelled (not bevelled), block (not bloc +/-mono), calorific (not caloric), favouring (not favoring), minimise (not minimize), immobilisation (not immobilization), specialised (not specialized), summarised (not summarized), characterise (not characterize), labelled (not labelled), analyse (not analyse), scrutinised (not scrutinized), devitalisation (not devitalization)
- Specific line corrections suggested by line are:
The current Part I:
145: oscillating, not oscillation
193: remove on after minimise
311: change what to so
348: remove of course
413-16: has there ever been a comparison of single MWP segment cost/performance to single IMF screw use?
460: replace what with which
724: self-cutting screws are more commonly termed self-drilling
742: change establishing: to namely:
760: start with Due to suspected study bias
762: change to as follows at the end of the sentence
769: add use after screw
780: change to and slow application, at end
794: change incidents to incidence
References: 1,2,7,8,12,13,14,18,19,5,58,74 – have all of these been translated into English for all authors to read/use and refer to? and are translated versions available for readers should they wish to access them?
To correct a change in format to references 37,39
What is a FAMI screw in reference 57??
Lastly, can a reference to the submitted but not yet accepted part II, be used in part I?
The current part II:
Introduction: can a reference to the submitted but not yet accepted part I, be used in part II?
This point also applies to lines 902-4, 1113, 1726
142-3: change likewise to like
174: place a gap between devices and in
Why are there separate supplements, if all of the text from these is already included in the body of the article? I suggest one or the other
234: change countersunk to countersink
261: change to lean to leaning of
270: change embodiment to arch bar
Page 13 onwards:
24-5: is 50% perforation referring to length of screw in root or width of root involved?
36: did any root perforations result in Root Canal Treatment or extraction? Plus 1594
81: change in succession of to after
154: change mandible to mandibular
239: remove 0 after indications
277: change remainder to remaining
297: change allowing to allow
359: add of after application
391: change resumed to reviewed
449: remove that, after so
450: change withdraw to withdrew, add as it after ambiguity, change repeating to repeated
451: change leaving to left
453: change past-time to previous
538: change of to the
548: change damages to damage sites
559: change for to of
564: change need to needed
565: change gingival to gingivae, remove of before compromised
573: change of, at end, to for
608: what is MFF? Is it MMF?
614: remove a
629: change consonant to consistent or coincident
646: change in- to incisions
659: add is after system and….
672: change has to have
762: what is OP? is it operative?
908-11: all text part of same sentence and no need for capital letters on patients and detail
1054: change bend to bent
1145: change does to do
1146: put which after threads
1182: change Walter to Wolter
1201: add leading after plate
1202: change the start of the sentence to with the, add may after height
1216: change “recommendable to” to “recommend that” and add is obtained, after situations
1349: change fractures to fracture
1402: add “due” before “to clamping”
1415: change inserting to insertion
1435: change what to which, add that it can also take them out of reach of cost restrained public funded health care systems
1447: change assumingly is, to is assumed to be:
1451: add general before anesthesia and change anesthesia to anaesthesia
1463: change as to like
1464: change as to like
1482: put space between demographic and variables
1517: put and after thick
1536: is it correct to use “long-term” when screws in this context are short-term use?
1544: should not simply be relied upon needs to be moved to the end of the sentence
1553: change cement to cementum
1554: change dentin to dentine
1560: change damages to damage
1627: change to heterogeneous
1670: change conceptionally to in concept
1702: change to had to rule out
1714: change assumed to used
1753: change connotes to refers to
1793: change to hardware extensions and MMF hybrid type screw heads is evidenced in part, but is both technique and hardware dependent
1836: move hints to after unambiguous
1853-54: could the differences in patient outcomes also be attributed to the use of wire or elastic IMF? As there will be different amounts of possible jaw movement and opening with each
1887-88: might this be explained by the use of the hand-held fracture reduction technique?
1917: change resonance to, to, acceptance of
1935: change to biased, and to placed after the word inattention
1939: remove resulting
1946: change to on after impact
1953: remove a after establishing
1954: put of after component
1955: change solutions to devices
1959: put simple after during
1969: put on after impacts
References: 11,12,13,14,15,16,44,46,48 – all have their format changed and just needs to be made the same as the other references
Has reference 55 been translated like in part I?
2326: is it meant to be intentional or unintentional damage with mini screw implants in this reference?
Reference 115; can this be used as a reference if the article is still in preparation and not accepted yet?
Comments on the Quality of English LanguageSee above
Reviewer 2 Report
Comments and Suggestions for Authors
I have been asked to review Part 1 and part 2 of this subject. This review covers both manuscripts.
I can see the amount of hard work that has gone into developing these two manuscripts with Part 1 discussing the new Omega shaped Wave Plate and its origins and part 2 looking at the history of Intermaxillary Fixation. It is an important and relevant subject. I appreciate there is no word count for the journal, but I feel both manuscripts are too long to keep the reader interested. Part 2 is 67 pages! You begin to loose what the take home message is.
Looking at the radiographic images I’m not sure what the purpose of the IMF is in the post operative period? In Part 1 with Figure 16 and part 2 Figures 2, 4 & 5 all are post op x rays where all fracture sites have be reduced and fixated with rigid plates yet the IMF is still present. What benefit does the IMF have post operatively when rigid plates are in place?
The discussion on MWF fixation is repeated again in Part 2 when it’s already been discussed in part 1
I feel that the same messages can be made with half the word content in both parts and a significant of images can be removed too.
Round 2
Reviewer 2 Report
Comments and Suggestions for Authors
The manuscript has been curtailed accordingly. However, I'm still concerned about the use of IMF when rigid plating has been applied. A comment needs to be made regarding this
Author Response
Dear Reviewer 2,
Thanks a lot for your new comment.
We understand that you would like to see a detailed comment included in the manuscript. Therefore we have added a whole section to the 4. Discussion of manuscript Part II about Post Operative MMF after ORIF.
